# The Status of Wildlife Damage Compensation in China

**DOI:** 10.3390/ani14020292

**Published:** 2024-01-17

**Authors:** Wenxia Wang, Torsten Wronski, Liangliang Yang

**Affiliations:** 1Research Institute of Forestry Policy and Information, Chinese Academy of Forestry, Beijing 100091, China; wang_wen_xia@163.com; 2School of Biological and Environmental Sciences, Liverpool John Moores University, James Parsons Building, Byrom Street, Liverpool L3 3AF, UK; 3Ecology and Nature Conservation Institute, Chinese Academy of Forestry, Beijing 100091, China; 4Key Laboratory of Biodiversity Conservation of National Forestry and Grassland Administration, Beijing 100091, China

**Keywords:** human–wildlife conflict, wildlife damage management, nuisance species, environmental justice

## Abstract

**Simple Summary:**

With the remarkable progress in wildlife conservation in China in recent years, the problem of human–wildlife conflict has notably increased. To appropriately resolve human–wildlife conflict and safeguard environmental justice, efficient wildlife damage compensation is crucial. This review attempts to compile information on the current state of wildlife damage compensation in China, describing the characteristics of wildlife damage and highlighting the necessity for compensation, regulation, and the management of wildlife damage compensation. We further compiled a list of nuisance species, and we exemplify detailed management practices in four case studies. Finally, we carefully evaluated the difficulties and challenges faced by China’s wildlife damage compensation schemes and provided recommendations for the future.

**Abstract:**

The conservation management of natural ecosystems in China has significantly improved in recent decades, resulting in the effective protection of wildlife and the restoration of habitats. With the rapid growth in wildlife populations and corresponding range expansions, incidents of human–wildlife conflict have notably increased across China. However, only a few studies have paid adequate attention to wildlife damage management and compensation. In our review, we focus on the foremost mitigation measure to combat human–wildlife conflict, i.e., compensation for damage caused by wildlife. We conducted a questionnaire survey and an in-depth review of the literature across 19 Chinese provinces and autonomous regions, resulting in a total of 78 relevant sources. We first introduce the concept of wildlife damage compensation in China, followed by a review of Chinese legislation and policies regarding wildlife damage compensation. We compiled a comprehensive list of nuisance species, and we showcase four case studies in which we exemplarily describe the current situation of wildlife damage compensation. We reflect on difficulties and challenges such as delayed damage assessments or compensation quotas that do not correspond to current market prices. Since local legislation is often insufficient or completely absent, we make suggestions on how to improve existing policies and compensation mechanisms.

## 1. Introduction

Since the dawn of industrialization, human population size has increased sevenfold [1], consequently resulting in a growing number of human–wildlife conflicts worldwide [2]. Human–wildlife conflicts have become a globally recognized problem mainly due to crop damage, predation on domestic livestock, or attacks on humans and their property [2,3,4,5]. To date, human–wildlife conflicts have come to be considered a significant threat to both biodiversity conservation and the economic growth of local communities [6,7,8,9,10,11,12]. One approach—among others (i.e., lethal control [8,13,14] or nonlethal or preventive control: fencing, livestock corrals, guard animals, or actively repelling wild animals [15,16])—to prevent human–wildlife conflict, is to alleviate the negative impact of wildlife damage through the provision of compensation to human victims [17,18,19], either through direct reimbursements or non-cash benefits such as the replacement of lost animals or the provision of food supplies [20]. This management tool was designed to provide environmental justice and to effectively protect the legitimate interests of communities (i.e., to recoup their losses), while at the same time preserving wildlife through increased levels of tolerance and by creating a positive attitude towards wildlife [3,21,22,23,24]. However, this is not always the outcome since generous compensation can lead to cutbacks of non-lethal prevention measures and, subsequently, to increased returns from agriculture, signifying an unintended subsidy for crop and livestock production [25].

Chinese hunting law strictly regulates hunting activities, requiring licenses for legal hunting. The law designates protected species and prohibits hunting during their breeding seasons. Violations result in fines, imprisonment, or confiscation of hunting tools [26,27]. Chinese conservation efforts to preserve biodiversity and control the impact of hunting on wildlife populations, has resulted in the effective protection of—often endangered—wildlife species and their habitats [26,27,28]. With the rapid growth of wildlife populations and corresponding range expansions, human–wildlife conflicts have notably increased across China [29,30,31,32]. In particular, the depredation of crops and domestic livestock, but also through the transmission of diseases and direct threats to human safety have become increasingly disastrous, resulting in serious economic losses and increasing grief among local communities [32,33,34,35,36,37,38,39,40]. Hence, this matter has become a focus for local and regional politicians, who have come to recognize the issue’s socioeconomic importance, particularly in provinces with rich biodiversity (e.g., Sichuan, Yunnan, Guizhou, Jilin, and Tibet) [41,42], and where compensation programs for local communities have already been implemented [30,43,44]. However, China’s compensation programs are met with numerous shortcomings and drawbacks, including the non-implementation of prevention measures, the failure to provide timely payments, financial unsustainability, and the absence of methods to assess success or failure [36,41,45].

In 2021, the China Forestry and Grassland Administration explicitly requested all provinces to actively formulate a judicious, practical, and science-based management scheme for wildlife damage compensation, providing for the protection of wildlife in accordance with Chinese law and safeguarding the legitimate rights and interests of victims. At present, 14 provinces and municipalities in China have implemented wildlife damage compensation management measures for damage occurring in terrestrial or wetland habitats, specifically those with high incidences of human–wildlife conflict and/or high abundances of potentially harmful wildlife species (hereafter termed nuisance species) [46]. Lately, only a few English studies have paid attention to the Wildlife Protection Law of China and the regulations and management of wildlife damage compensation therein [26,45,47]. In our article, we therefore focus on the foremost mitigation measure to combat human–wildlife conflict, i.e., the compensation of damage caused by terrestrial wildlife species, aiming to (i) introduce the concept of wildlife damage compensation in China, (ii) comprehensively review Chinese legislation and policies regarding wildlife damage compensation, and (iii) compile a comprehensive list of nuisance species in China. Furthermore, (iv) we present four case studies, to describe the current state of wildlife damage and compensation in detail and (v) we make suggestions on how to improve existing policies and compensation mechanisms in the country.

## 2. Materials and Methods

In a first step, four online search engines, i.e., Google Scholar, CNKI (Chinese National Knowledge Infrastructure), Microsoft Bing, and Baidu, were used to search six search terms (and combinations thereof), i.e., wildlife damage, human–wildlife conflict, crop damage, livestock predation, wildlife damage mitigation, and compensation. The search was filtered by country (China), date (2001 to 2023), language (Chinese and English), and sorted by relevance. The first 100 results (maximum) per each search term were saved and screened to remove initial duplicates. Using Endnote version X9 (Clarivate^TM^, Clarivate PLC, London, UK), the results were checked for further duplications, and screened by title and abstract to omit marine-, non-Chinese-, non-conservation-, and non-wildlife-focused studies. Subsequently, all scientific journal articles, scholarly and government publications, and internet sources, such as local and national news pages, were collated. A full text search was then performed to proof eligibility for data collection, i.e., eliminating results missed by previous filters, such as information lacking relevance, or studies that were not focused on terrestrial wildlife damage compensation in China. In the end, the data presented in this study resulted from 78 sources (56 Chinese, 22 English) from 19 Chinese provinces and autonomous regions, including Yunnan, Shanxi, Shaanxi, Jilin, Gansu, Xizang (Tibet), Qinghai, Anhui, Guizhou, Heilongjiang, Inner Mongolia, Sichuan, Liaoning, Guangxi, Xinjiang, Hunan, and Hainan, as well as two local municipalities, i.e., Beijing and Tianjin City (Figure 1). From the remaining sources, we extracted information on nuisance species involved in compensation programs, the time, type, and severity of the damage (Appendix A). In addition, we surveyed Chinese legislation and policies relating to wildlife damage compensation at the national and local level, mainly based on the ‘Wildlife Protection Law of the People’s Republic of China’ as well as on local wildlife damage compensation regulations from 14 Chinese provinces and autonomous regions (Appendix A). 

Secondly, from June to October 2021, we conducted an interview survey on wildlife damage incidents and subsequent government compensations in Hainan Province and Tianjin municipality [48,49]. A total of 3600 and 45 interview questionnaires were sent to Hainan and Tianjin, respectively, while 3597 and 40 valid responses were returned. The interviewees included were mainly local farmers, livestock keepers, managers, and rangers working in protected areas or forest plantations, as well as the leaders of local forest authorities. We requested information on nuisance species involved in compensation programs, the time, type, and severity of the damage (Appendix A). Respondents were also asked to provide information and suggestions on how to improve wildlife damage management and compensation mechanisms.

## 3. Characteristics of Wildlife Damage in China and the Necessity for Compensation

### 3.1. Characteristics of Wildlife Damage

Wildlife damage in China refers mainly to (1) damage to local (subsistence) and cash (commercial) crops, stored food, timber, or personal property; (2) threats to human health and safety via disease transmission or direct impact, (3) threats to the safety and welfare of livestock through predation and/or disease transmission, and (4) general nuisances caused by the noise or other activities of wild animals [12,50]. Three reasons for the recent increase in wildlife damage cases in China were proposed: (1) China made considerable progress in the conservation of its biological diversity, resulting in the sustained protection of wildlife and the expansion of its ranges and activities, while at the same time poaching, i.e., the illegal hunting of wildlife, became less prominent [27]. (2) The continuous expansion of human activities, such as agriculture and urbanization, have encroached on the habitats of wild animals. Although China has established many protected areas to constrain the impact of anthropogenic activities, the development of urban and peri-urban areas has inevitably led to overlaps between wildlife, humans, and their livestock, and thus to more human–wildlife conflicts. (3) Due to a lack of natural predators, the populations of some wildlife species, such as wild boar (*Sus scrofa*) and some rodents, have risen to unsustainable levels [51,52].

The threats wildlife damage can pose to human property or safety are manifold, depending on the size, diet, or affinity to humans of the wildlife involved. For example, different species of wildlife have different biological characteristics, thus involving different degrees of damage severity or acuteness and thus requiring different responses from law enforcement. Many wild animals are aggressive towards humans, especially large mammals can cause serious injuries or even fatalities [40,53]. By contrast, small animals are less dangerous but more abundant, creating serious crop damage which can lead to famines and consequently to human hardships [45,54]. Moreover, different geographic regions or habitats are inhabited by different wildlife species. For example, Asian elephants (*Elaphus maximus*) and wild boar are frequent nuisance species in forest habitats [55], while wild yaks (*Bos mutus*) or kiangs (*Equus kiang*) compete with livestock for pasture in grassland habitats [56]. In terrestrial habitats, mammals usually instigate the harm, while in wetland habitats migratory birds are the prevailing nuisance species [49,57]. Lastly, although wild animals are the cause of wildlife damage, they are not eligible civil subjects according to the ‘Civil Code of Law of the People’s Republic of China’. Wild animals cannot be the subject of compensation requests, instead they are considered state property and victims of wildlife damage need to render their damage compensation requests to local governments.

### 3.2. Necessity of Compensation 

Compensation for wildlife damage is considered a key component to mitigate human wildlife conflict around the world, particularly important for communities living in high conflict areas [40,58,59,60,61,62,63]. Most compensation schemes take retroactive action (i.e., after the incident has occurred), through the payment of compensation based on the estimation of the actual damage—either through direct reimbursements or by non-cash benefits [4,58,59,61,62,63]. 

In China wildlife damage compensation requests are usually directed towards the local or district governments and their liable departments, entitled to provide economic alleviations in accordance with regional laws and regulations [12,55,64,65]. Wildlife damage victims are individuals or organizations suffering personal or property damage and are therefore eligible to file requests for compensation. Compensation payments are mostly requested for the loss of livestock, for medical expenses or disability benefits, or for families facing the fatality of a family member. Compensation payments are not only applied to mitigate and reduce the economic burden of the individual (or organization), but also to improve and increase the community’s tolerance towards wildlife and therefore to reduce retaliatory actions [35,66]. The necessity of wildlife damage compensation in China rests on two major pillars:(1)The need to protect the legitimate rights and interests of victims (environmental justice)

Many large wildlife species have strong capabilities to attack and injure a person, instigating serious consequences for the victim and their family. By law, local governments and social organizations are expected to increase the availability and transparency of wildlife damage compensation schemes and enable victims to build and strengthen their awareness of their right to protect their legitimate rights and interests. The victims should be empowered and qualified to master the provisions and procedures of applying for wildlife damage compensation, and to ensure that their lawful rights and interests are effectively protected [12,67]. Wildlife damage mostly occurs in remote rural areas, where most residents have no professional guidance on how to appropriately respond to the impairment. Instead of applying for compensation for the encountered crop or property damage, residents rather try to displace or even kill wildlife in retaliation for the damage. Governmental guidance should inform victims on how to correctly respond and how to submit a compensation request to the local authorities and encourage them to do so. For victims, human–wildlife conflict is often a serious issue, causing sustained opposition towards governmental conservation efforts. However, international and national experiences have shown that if moderate and timely compensation is provided, the victims of wildlife damage are willing to abandon retaliation and apply for reimbursement to the respective compensation scheme [47,68]. Advocates of compensation argue that compensation schemes increase tolerance towards wildlife, decrease retaliatory killings, and help build community support for the conservation of wildlife [55,61,66,69,70]. By contrast, long-term exposure to the risk of wildlife-related injury or death and untimely compensation can increase hostility towards wildlife and lead to more incidents of retaliatory killing, particularly among rural communities living near protected areas [60,71].

(2)The need to maintain social peace and stability

Damage caused by wildlife, including crop raiding, disease transmission, injury to humans, and livestock depredation, are key drivers of negative interactions between people and wildlife [72]. Wildlife damage can seriously affect a community’s well-being and safety, in the worst case leading to social instability [9,67]. For instance, Asian elephants in Xishuangbanna Municipality regularly cause serious damage to local crops and property (see Case study 1) [67]. Consequently, farmers have developed strong negative attitudes, specifically towards elephants but also towards wild boar, when agricultural products have been raided [53]. To mitigate these negative consequences for the community’s social stability, the local government realized that it is imperative to appropriately and in a timely manner provide legal compensation, to pacify and resolve local disputes, and to thus maintain social harmony and stability. Local authorities in Xishuangbanna Municipality also understood that proactive precautions, i.e., wildlife damage prevention measures, would help local farmers to reduce the frequency and degree of damage and subsequently could act to diminish environmental injustice.

## 4. Regulation and Management of Wildlife Damage Compensation in China 

### 4.1. National Legislation

Currently, China does not have a specific law regulating compensation for wildlife damage. Instead, the management of wildlife damage compensation is stipulated in Article 18 and 19 of the ‘Wildlife Protection Law of the People’s Republic of China’ [22]. Article 18 requires that the relevant local governments must take measures to prevent and control damage caused by wildlife and ensure the safety of people, their livestock, and agricultural products. Article 19 stipulates that where casualties or property losses are caused by wild animals protected by this law, compensation shall be offered by the local government. This applies particularly—but not only—to damage caused by wildlife under ‘special state protection’ listed under Article 10 of the Wildlife Protection Law. Article 19 further states that prevention and compensation measures shall be formulated by the governments of provinces, autonomous regions, and municipalities. The relevant local governments are encouraged to establish communal insurance schemes which provide funds for wildlife damage compensation. Such funds shall be subsidized by central finance, following the relevant provisions of the national government. Such subsidies are useful tools for local governments when implementing relevant damage prevention and control measures, or when offering compensation contracts [22]. 

### 4.2. Local Legislation 

Yunnan was the first Chinese province to formulate compensation management measures for wildlife damage in 1998. By 2008, the China National Forestry and Grassland Administration (formerly the State Forestry Administration) decided to start a nationwide pilot program for compensating wildlife damage in Jilin, Yunnan, Shaanxi, and Xizang Provinces, i.e., provinces that had already established local compensation measures and that could act as pilot areas. The local governments and administrative forestry departments in these pilot areas adopted the proposed legislation and used the subsidies of central finance to implement compensation schemes and primarily to compensate for losses caused by national key protected wildlife species [22,56]. Today, 14 provinces, autonomous regions, and municipalities in China have formulated compensation measures for wildlife damage, including Yunnan, Shaanxi, Jilin, Beijing, Gansu, Xizang, Qinghai, Anhui, Guizhou, and Heilongjiang (for details see Appendix A). Most recently, i.e., in 2023, Shanxi, Inner Mongolia, Sichuan, and Liaoning Provinces implemented wildlife damage compensation schemes, while Hainan Province and Tianjin Municipality are currently defining wildlife damage compensation measures [73,74,75,76,77,78,79,80,81,82,83,84,85,86]. 

#### 4.2.1. Main Subjects of Current Compensation Schemes 

China’s local wildlife damage compensation schemes include the following four sections: Explain the purpose, basis, and scope of the compensation measures;Stipulate the application and exclusion of compensations for wildlife damage;Define the procedures and requirements for the application, investigation, identification, and verification of wildlife damage and inform local communities about these procedures;Make provisions regarding the legal liability for wildlife damage.

#### 4.2.2. Comparability of Compensation Standards 

Specific compensation standards and management procedures vary greatly between the 14 provinces and autonomous regions (for details see Appendix A). However, following the general guidelines provided by the ‘Wildlife Protection Law of the People’s Republic of China’, wildlife damage is roughly divided into personal injury and property damage, comprising five generally applicable wildlife damage compensation categories:Compensation for personal injury causing partial loss of labor force will be 2 to 15 times the local average income of the previous year, compensation for total loss of labor force will be 8 to 25 times the local average income of the previous year, while compensation for death will be 10 to 30 times the local average income of the previous year;Compensation for crop- or economic forest-damage will be 50 to 80% of the actual loss;Compensation for livestock or poultry injuries will be 50 to 70% of the treatment costs;Compensation for the loss of livestock or poultry will be 50% to 100% of the average market price;Compensation for repairing damaged legal property is supposed to be 70% of the repair fee, while non-repairable property will be compensated with 50% of the average market price.

## 5. Nuisance Species in China

The type and extent of wildlife damage varies greatly between the different biogeographic regions of China [87], depending on the human population density, the climate zone and the corresponding habitat types [33,35]. Areas with a high and severe degree of wildlife damage are usually close to important wildlife habitats, protected areas, or migration routes of nomadic species [35,55,56]. Certain vegetation and habitat types link to specific nuisance species and the kind of damage they cause. While predators usually cause injury or death to livestock and poultry, herbivores compete with domestic livestock for pasture or cause damage to or total loss of harvests and property. A comprehensive list of nuisance species reported in 19 Chinese provinces, autonomous regions, and municipalities is provided in Appendix A [24,31,32,33,36,37,48,49,54,55,56,65,88,89,90,91,92,93,94,95,96,97,98,99,100,101,102,103,104,105,106,107,108,109,110,111,112,113,114,115,116,117,118,119,120].

In total we recorded 185 nuisance species, comprising 132 mammals, 44 birds, and 9 reptile species. Wildlife species involved in human–wildlife conflict vary greatly between the seven major biogeographic regions of China (Sun et al., 2020) [87], with Hainan having the highest count of nuisance species (22), followed by Hunan (19), Sichuan (18), Tianjin (16), Yunnan (16), Jilin (15), and Xizang (13). Tianjin and Hainan showed the highest number of birds involved, with 13 and 12 species, respectively, while Sichuan (17), Yunnan (13), Hunan (13), and Xizang (13) showed the highest number of mammal-induced incidences. Damage caused by migratory birds is particularly high in the coastal wetlands of Tianjin Municipality and Liaoning Province, where migratory birds consume large amounts of farmed fish, shrimps, and crabs [57]. Compared to other regions covered by our survey (Appendix A), the proportion of incidences related to large and medium-sized mammals is relatively higher in the northeast (Siberian tiger, *Panthera tigris altaica*; Asiatic black bear, *Ursus thibetanus*; and Amur leopard, *Panthera pardus orientalis*), the northwest (wolf, *Canis lupus*; brown bear, *Ursus arctos*; and snow leopard, *Panthera uncia*), and in the south of China (Asian elephant; Indochinese leopard, *Panthera pardus delacouri*; Asiatic black bear; and wild boar). 

Human–wildlife conflict with large mammals occurs mainly in or near nature reserves, as well as in remote mountain areas with a relatively impoverished human population [47]. For example, Hunchun in Jilin Province is a typical area suffering wild boar damage but is also an important region for the conservation of Siberian tiger [53]. Another remote area with a high degree of human–carnivore conflict is the Taxkorgan Nature Reserve in the Pamir and Karkorum Mountains of northwestern China. Here, mainly wolf and dhole (*Cuon alpinus*) were responsible for a high incidence of livestock predation, with 127 reported cases and a total of 583 animals killed from 2011 to 2013 [37]. The survey further revealed that snow leopard, wolf, Eurasian lynx (*Lynx lynx*), and brown bear were the major livestock predators in Qilianshan National Nature Reserve, in Gansu Province [34], as well as in the Qomolangma National Nature Reserve, near Mt. Everest in the Tibet Autonomous Region [35]. Similarly, to the situation reported by [35] for predators in Gansu and Tibet, the North China leopard (*Panthera pardus japonensis*) is facing the challenge of decreasing prey numbers in the Tieqiaoshan Provincial Nature Reserve, Shanxi Province. Due to the low numbers of natural prey animals in the reserve, leopards have been reported increasingly prey on livestock in and around the protected area, negatively impacting the local communities who entirely depend on livestock rearing as a source of income [38]. Human casualties were mainly reported in Yunnan Province, where incidences with wild Asian elephants lead to injury or the deaths of local people or to the devastation of local cash crops [43,121]. Apart from periodic incidences with elephants, the Asiatic black bear is the most conflict-prone mammal species around the Daxueshan Nature Reserve, followed by the rhesus macaque (*Macaca mulatta*) and the South Asian sambar (*Rusa unicolor*) [36]. The major crop raiders in China are primates, especially the rhesus macaque which is widespread in Hunan, Hainan, and Guangxi Provinces and which is frequently reported destroying large quantities of crops [32,48,96]. Even in rural Beijing, frequent and severe crop damage has been reported, mainly caused by macaques, wild boar, and tolai hare (*Lepus tolai*) [31,88].

Overall, our survey results suggest that wildlife damage in China has been underestimated. The questionnaire surveys conducted in Tianjin Municipality and Hainan Province [48,49] found more nuisance species than expected based on previous records. Given that Hainan and Tianjin have comparatively few nuisance species and relatively low rates of human–wildlife conflict, it can be expected that the underestimation of human–wildlife conflict in other areas is far more serious than is currently acknowledged. 

## 6. Case Studies

Although China has been compensating wildlife damage for more than 30 years, detailed information—especially from official sources—is scarce. Most research on wildlife damage and the provisioning of compensation for human victims have been carried out in Yunnan, Tibet, and Jilin Provinces as well as in Beijing’s Miyun District (Figure 1). Herein, we report four case studies from these regions, reviewing open-access research publications [11,56,65,116,122] that are readily available to enable study of the state of wildlife damage and compensation in more detail.

### 6.1. Yunnan Province

Yunnan Province is particularly rich in natural resources and has extraordinary biodiversity, including the only two remaining population of Asian elephants in China. One population occurs in the Nangunhe National Nature Reserve, near the city of Lingcang in the west of Yunan, the second population occurs in the mountains around the city of Jinghong in the south of Yunan, where it persists in several small nature reserves such as the Xishuangbanna or the Napanhe National Nature Reserves. Yunnan Province traditionally reports the highest numbers of wildlife damage incidents in China and has therefore the most developed and best-documented compensation schemes [11,56,122].

The Nangunhe National Nature Reserve, with an area of 508.8 km^2^, comprises a nearly pristine habitat with functional ecological interactions and outstanding tropical diversity [56]. However, the continuous growth in the human population, intensified farming around the reserve, and deforestation for rubber plantations have resulted in a reduction in suitable wildlife habitats, increased crop raiding, and thus an aggravated conflict between humans and wild animals. Besides the Asian elephant, the main nuisance species in the reserve are Asiatic black bear, wild boar, wolf, rhesus macaque, and leopard (Appendix A) [56]. The main types of damage in 2018 were local crops (33.92 km^2^—crop and timber damage are traditionally given in ‘mu’, a Chinese unit for area), timber (2130 m^2^), livestock (34 individuals), and six human casualties. In the same year, the total amount of compensation paid for wildlife damage around the reserve was 1.1m Yuan (153.1k USD), including Nuoliang Township (19.6k Yuan/2.7k USD), Shan Jia Township (35.9k Yuan/5.0k USD), Mengdong Township (44.3k Yuan/6.2k USD), Menglai Township (52.6k Yuan/7.4k USD), Mengjiao Township (57.7k Yuan/8.1k USD), Ban Lao Township (232.0k Yuan/32.5k USD), and Banhong Township (641.6k Yuan/89.8k USD) [56]. The most damaging nuisance species around the reserve is the wild boar, with compensation accounting for more than 80% of the total amount [56]. In recent years, the number of wild animals in the area has increased significantly, which is a major reason for the increase in wildlife damage incidents. With improving economic development, the prices of agricultural products have increased, and local people have complained that the status quo of wildlife damage compensation is insufficient. Moreover, due to a lack of specific funds, it is often difficult to carry out appropriate investigations, to collect evidence and report wildlife damage to governmental authorities. There is also little community outreach and education related to local wildlife damage, and thus public awareness of active defense mechanisms and adequate prevention is low. As a result, the tolerance towards crop raiding and wildlife accidents is minimal, and the enthusiasm to participate in wildlife conservation is dwindling [56].

The city of Jinghong, located in the center of Xishuangbanna Dai Autonomous Prefecture, is surrounded by several small, protected areas, comprising 13.00 km^2^ of national and 441 km^2^ of municipal nature reserves. Eighty-five percent of the prefecture is still covered by forest which represents the main habitat of wild Asian elephants in China. Due to an ever-increasing loss of ancestral habitat, elephants are obliged to use crops, rather than their natural diet, as their main food source [123]. Agricultural products have good palatability, a high nutritional value, and are therefore easy to digest. Consequently, the local communities in the prefecture are facing profound damage to local and cash crops. Moreover, using their traditional migration routes, elephants pass through rubber and tea plantations, where they cause considerable damage [122]. Other conflict-prone wildlife species in the reserves include wild boar, Asiatic black bear, rhesus macaque, some ungulates, and several venomous snakes. The main types of damage involve local crops, cash crops (e.g., mainly rubber, tea, and coffee), timber, livestock, and human casualties. From 1990 to 2017, wildlife damage around Jinghong caused a total economic loss of 240.4m Yuan (33.7m USD), 121 casualties (including 14 fatalities), and de facto compensation of 27.4m Yuan (3.8m USD), corresponding to a compensation rate of only 11.4% [11]. For the period from 1990 to 2009, this means that wildlife damaged 32,182 tons of local crops, 11.9k tons of cash crops, 2.6m timber trees, 5189 pieces of livestock and poultry, and 70 human casualties (including eight fatalities) [11]. In 2010, the Xishuangbanna Forestry Bureau signed a contract with the Xishuangbanna Branch of the ‘China Pacific Property Insurance’, to establish and implement public wildlife liability insurance. The insurance company is responsible for the settlement of claims, while the forestry department assists in conducting site investigations and damage assessment and verification. After the implementation of wildlife liability insurance, i.e., the period from 2011 to 2017, Song et al. (2019) [11] reported 1737 human–wildlife incidents with Asian elephants (and other wild animals), comprising 51 human casualties (including 6 fatalities), a direct economic loss of 646.9m Yuan (90.6m USD), and insurance claims of 15.7m Yuan (2.2m USD).

### 6.2. Tibet Autonomous Region

Tibet (or Xizang) is one of the richest and most compelling areas in terms of biodiversity in China—and probably worldwide—having seen significant improvement in nature conservation in recent decades. Large areas of alpine grasslands were assigned protected area status (e.g., Mangkang *Rhinopithecus bieti* National Nature Reserve [1853 km^2^], Qiangtang National Nature Reserve [247,120 km^2^], or Lhalu Wetland National Nature Reserve [12.2 km^2^]), resulting in the rapid recovery of wildlife populations [124]. In particular, the growing populations of Tibetan antelope (*Pantholops hodgsonii*), Tibetan gazelle (*Procapra picticaudata*), kiang, white-lipped deer (*Przewalskium albirostris*), and wild yaks have resulted in increased competition for pastures with domestic livestock, while growing numbers of predators such as wolf, brown bear, Eurasian lynx, or snow leopard have carried out recurrent attacks on domestic livestock (mostly sheep, goat, and domestic yak), and seriously affected the livelihoods of local livestock keepers. In their study on wildlife damage compensation incidents in Tibet (2007 to 2019), Liang et al. (2020), identified the cause and type of damage (Appendix A) and quantified the economic losses linked to each type. According to the 2018 data, the annual losses in Tibet amounted to 78.3m Yuan (10.9m USD), including 59.1m Yuan (8.3m USD) for livestock casualties, 495.6k Yuan (69.4k USD) for damage to housing and property, 901.2k Yuan (126.2k USD) for local and cash crop raiding, and 1.6m Yuan (0.2m USD) for human disabilities and fatalities. During the study period, the cumulative compensation for wildlife damage in Tibet was about 960m Yuan (130m USD), gradually rising from 11.5m Yuan (1.6m USD) in 2007 to 119m Yuan in 2019 (16.6m USD; [56]).

From 2007 to 2015, Tibet’s wildlife damage compensation funds were jointly raised by the autonomous region, cities, and municipalities. In 2016, the government of Tibet transferred the management of wildlife damage compensation to private insurance companies—executed via government procurement services, meaning that insurers were liable for their financial benefits or losses [56]. At present, compensation for wildlife damage in Tibet is uniformly insured by the local Forestry and Grassland Bureaus, acting as a legal person to ensure the effective implementation of the wildlife damage insurance scheme. Despite this progress, there are noteworthy problems related to the implementation of the wildlife damage compensation scheme [56,104]. For example, pastures gnawed by wild herbivores are difficult to assess and to be compensated for, leading to frustrated and dissatisfied livestock keepers, who become unwilling to protect wildlife and instead to matters into their own hands and retaliate. The compensation standards for casualties among domestic livestock are relatively low, and a timely verification and assessment of damage is difficult due to the remoteness of many areas. Consequently, this has negative effects on the timely compensation of victims and their attitude towards wildlife. Moreover, compensation standards for the assessment of lost or damaged crops and timber need to be further refined and adjusted by local lawmakers. 

### 6.3. Jilin Province

Jilin Province, located in northeast China, is an important forestry province, with high forest coverage (45.2%) and abundant wildlife resources [125]. Wildlife is still widespread across the province but is most numerous in the Changbai Mountain National Nature Reserve located at the border to North Corea in the south of Jilin Province. The most prominent nuisance species are Siberian tiger, Amur leopard, Eurasian lynx, leopard cat (*Prionailurus bengalensis*), Asiatic black bear, wolf, Siberian weasel (*Mustela sibirica*), yellow-throated marten (*Martes flavigula*), badger (*Meles meles*), red deer (*Cervus elaphus*), and wild boar. The province is also rich in birdlife of which particularly birds of prey (*Falconiformes* and *Strigiformes*), geese, and ducks (*Anseriformes*) are considered nuisance species (Appendix A). In recent years, the number of wild animals in the province has increased significantly, due to a stringent hunting ban and strict protection measures [126]. In their study of wildlife damage compensation incidents in Jilin Province from 2007 to 2013, Sun et al. (2015) [65] found that the number of crop raiding incidents involving wild boar, as well as the number of human and livestock casualties—instigated by large predators, such as Siberian tiger and black bear—significantly increased. According to Sun’s (2015) study [65], the main types of damage are local crop damage (maize, rice, and soya), livestock damage (goat and cattle) and personal injury, wherein crop damage accounts for more than 90% of all reported cases (see Appendix A). The wild boar is the most common nuisance species, accounting for 94.1% of the reported wildlife damage incidents. During the study period, the province received a total of 26,599 requests for wildlife damage compensation, with the number of incidents continuously increasing from 993 cases in 2007 to 6550 cases in 2013. The cumulative compensation for wildlife damage in Jilin Province over the past seven years was 89.8m Yuan (12.6m USD) [65]. The study further highlighted that the management of wildlife damage compensation cases in Jilin Province—including supervision, inspection, and verification—is relatively weak. The management of wildlife damage compensation data needs to be improved, and the performance and promotion of wildlife damage prevention measures need to be enhanced.

### 6.4. Beijing Miyun District

Miyun District, located in the northeast of Beijing Municipality, is an important source of drinking water for the capital of China [88]. Furthermore, the district was identified as a conservation and ecological development area, due to its nearly intact, natural habitat with rich forest resources and good living conditions for various wildlife species [88]. Many villages and small-scale farms in Miyun District are located near small, protected areas or woodlands, which inevitably leads to human–wildlife conflicts. The problem has been well known for a long time, prompting a study by Li Jian (2015), who analyzed the compensation of wildlife damage incidents in the region from 2009 to 2014. The nuisance species in Miyun District include mammals such as the tolai hare, wild boar, and badger causing the most serious damage to local crops, especially maize (Appendix A). Among the birds, Magpie (*Pica pica*), Azure-winged magpie (*Cyanopica cyana*), Eurasian tree sparrow (*Passer montanus*), Common pheasant (*Phasianus colchicus*), Eagle owl (*Bubo bubo*), and some other birds of prey are the main causes of human–wildlife conflict, mainly because of raiding on various crops and fruit trees or attacking poultry. From 2009 to 2014, a total of 5723 wildlife incidents were reported, with compensation worth of 4.7m Yuan (0.65m USD) handed out. Compensation has constantly increased from 347k Yuan (48.5k USD) in 2009 to 99k Yuan (139.4k USD) in 2011, 944k Yuan (131.9k USD) in 2012, 1.2m Yuan (167.8m USD) in 2013, and to 1.2m Yuan (166.1m USD) in 2014 (Li, 2015). Due to the rapid increase in wildlife numbers in recent years—mainly owing to improved conservation—and the ongoing reclamation of natural habitats for agriculture, human–wildlife conflicts in Miyun District have dramatically expanded, resulting in demands for better wildlife damage compensation management by national and local authorities [31,88]. Otherwise, the reduction in suitable habitats for wildlife will aggravate the conflict between humans and wild animals, eventually leading to retaliation against nuisance species and a negative attitude towards nature in general. 

## 7. Difficulties and Challenges 

The reasons for the failure of compensation schemes are numerous, including inadequate compensation, a lack of sustainable funding, and the creation of incentives detrimental to conservation, i.e., so-called ‘moral hazards’ such as the over-reporting of losses [127]. Reviewing China’s management of wildlife damage compensation, we identified the following main problems:(1)Local legislation on wildlife damage compensation is often insufficient or completely absent. Until 2023, only 14 provinces and autonomous regions had formulated specific compensation management measures (Appendix A).(2)Although the governments of these provinces have implemented compensation standards (Appendix A), there is still an urgent need to improve, align, and harmonize compensation schemes between provinces. Moreover, the existing compensation standards need to be more specific regarding the nature of damage or loss [56,128].(3)Challenges remain to accurately assess the magnitude of crop damage, especially that of cash crops, forest plantations, or aquaculture, since different wildlife species cause different degrees of damage at different growth stages [55].(4)Investigating and verifying the cause of wildlife damage remains difficult. Damaged pastures, for example, are often located in remote mountainous areas of considerable size, with insufficient transportation and poor infrastructure, making the verification of wildlife damage difficult, time consuming, and cost intensive. For instance, in Tibet, herbivorous wildlife species such as Himalayan marmots (*Marmota himalayana*), plateau pikas (*Ochotona curzoniae*), kiangs, or wild yaks compete for or trample pastures contracted by local herders for livestock grazing, ultimately resulting in severe economic losses that cannot be clearly assigned to one or other nuisance species [56,104,105].(5)Damage cannot be assessed in a timely manner, so that victims fail to receive compensation in time and may thus become dissatisfied with the legislation and commit acts of retaliation against the relevant wildlife species [55,56,64]. More qualified veterinarians are therefore needed to verify the cause of damage (or death). Examples from Europe demonstrate that the prompt availability of skilled professionals helps to avoid a waste of public resources for unjust compensation and the distortion of the genuine impact of the depredation [2,129]. Some officials have pointed out that the extent of livestock depredation was possibly overestimated because putative victims can fake evidence of damage to receive compensation. On the other hand, residents complain that qualified depredation evidence is hard to obtain and sometimes evidence is lost, resulting in an underestimation of actual damage [35].(6)Previously fixed compensation quotas showed large discrepancies compared to current market prices, increasing the dissatisfaction of aggrieved victims [12,56]. Given the actual damage and financial losses, compensation quotas are relatively low in China. Farmers in the Xishuangbanna Dai Autonomous Prefecture in Yunnan Province, for example, were broadly dissatisfied with the current insurance system, and their level of satisfaction was closely associated with the compensation quota, i.e., the percentage of lost rubber reimbursed by the insurance [127]. With the fast development of the economy, the price for rubber also increased rapidly, but the compensation quotas did not keep up.(7)The situation is further compounded by a general lack of compensation funds. With the increasing size of wildlife populations, a growth in wildlife damage incidences and the expansion of areas impacted by human–wildlife conflict go hand in hand. Since government compensation funds were established when population numbers were lower and conservation measures had not yet taken effect, today’s compensation funds are insufficient to cover the increased number of incidences. This development leads to constantly increasing insurance premiums, eventually resulting in the risk of undersupplied compensation funds in the future [12,33].(8)Finally, the approval of compensation must be closely tied to effective prevention measures. Only if the affected party can prove that such measures were in place, should compensation be approved. In addition, the local knowledge of wildlife damage prevention measures is often inadequate, and farmers lack the necessary education, training, and tools to apply for compensation [30,130]. Therefore, compensation for wildlife damage from governmental or commercial insurance should not only cover the de facto damage but should also include funds for prevention and coaching activities.

## 8. Recommendations

Given the above challenges, the following recommendations are proposed:Generally, adapting governmental restrictions (e.g., seasonal grazing limits), improving wildlife management practices in and around protected areas, and carefully designing compensation schemes should be reinforced with educational activities to increase awareness and support for the protection of wildlife and ecosystems [34,55,56,66].Wildlife damage compensation systems should be established and/or improved as soon as possible, and they should be adapted to the local state of wildlife damage and the social and economic development of the community. This includes improving and harmonizing management measures and standards for wildlife damage compensation across provinces, and gradually increasing the amount of compensation, to safeguard the interests of affected victims in the long-term.Financial compensation should be provided immediately or at least as soon as possible. We further advise that the amount of compensation should be based on the market value or yield of crops, not on a fixed government quota. This would be fairer and more easily accepted by local communities [24].The establishment of multi-stakeholder compensation mechanisms should be encouraged, and the formation of wildlife damage insurance systems should be promoted, combining commercial and legislative coverage to share the risk of damage and safeguard the interests and economic losses of victims. Wildlife damage public liability insurance is a form of compensation that is purchased by the government on behalf of the farmer or livestock keeper, and once wildlife damage occurs, the insurance company compensates the victim for personal and property losses [50,55,131]. Such public–private insurance models are gaining recognition and are widely accepted as an improvement to traditional, purely governmental compensation schemes [127]. The multi-stakeholder compensation scheme implemented by Yunnan and Qinghai Provinces as well as by the Xizang Autonomous Region can hereby serve as models for other provinces and municipalities. The government of Xizang Province has even gone a step further, suggesting that the current compensation program should be extended to insurance purchased by local governments to supplement standard compensation for the destruction of homes or the loss of livestock [40,56,116].Regular surveys of wildlife damage should be conducted to recognize potential nuisance species in a timely manner and to monitor their population development, activity patterns, behavior, dietary preferences, and key distribution areas [9,88]. Scientific population control plans for nuisance species that are suitable for population regulation should be formulated and implemented by professional wildlife ecologists, enabling sustainable control of increasing populations [95]. Culling activities must be carried out by professional governmental hunters, and benefits from culled animals should be transferred to neighboring local communities.Local ecological knowledge should be incorporated to develop innovative approaches to mitigate human–wildlife conflict, e.g., a tree planting initiative in Yunnan Province restored and improved elephant habitats, attempting to keep elephants away from plantations and human settlements [55]. This approach was made possible because the local government was actively enhancing wildlife damage prevention measures and publicly advertising such proactive approaches.The awareness of local communities of wildlife protection, prevention, and control measures should be increased. Wildlife protection laws and regulations should be actively publicized through directed promotion and education activities such as the distribution of pamphlets and the use of online multimedia platforms.Finally, local governments need to commit to wildlife conservation and enforce the existing law by cracking down on illegal and criminal activities such as hunting and the trade of wild animals and products thereof.

## 9. Conclusions

In conclusion, China’s wildlife damage compensation system reflects a commitment to balancing conservation and human–wildlife coexistence. The policy acknowledges the economic impact of wildlife damage on local communities and provides a framework for compensating losses. While efforts have been made to address conflicts, ongoing evaluation and refinement of the compensation mechanism are essential to ensure effective conservation measures and sustainable harmony between humans and wildlife.

## Figures and Tables

**Figure 1 animals-14-00292-f001:**
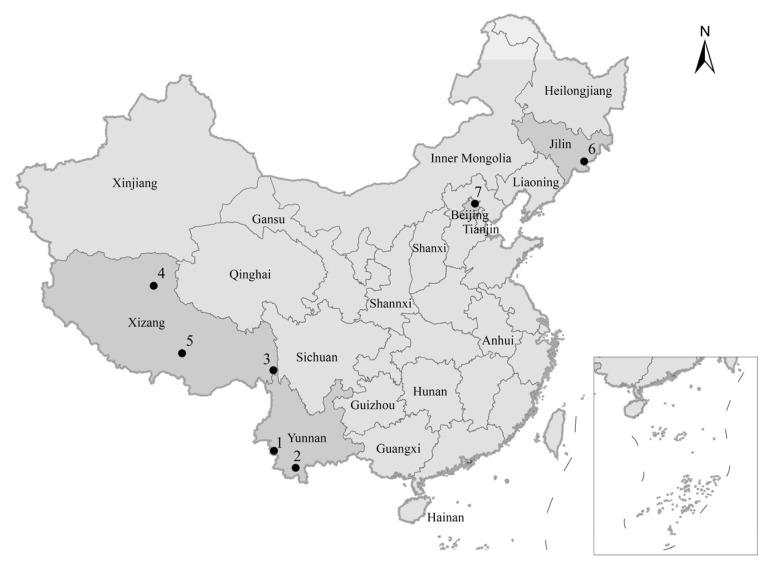
The Peoples Republic of China, the provinces, autonomous regions, and municipalities mentioned in the text (insert shows South China Sea). Dots indicate the locations of protected areas mentioned in the case studies: 1. Nangunhe National Nature Reserve, 2. the city of Jinghong, 3. Mangkang *Rhinopithecus bieti* National Nature Reserve, 4. Qiangtang National Nature Reserve, 5. Lhalu Wetland National Nature Reserve, 6. Changbai Mountain National Nature Reserve, 7. Beijing Miyun District.

## Data Availability

Data are contained within the article and Appendix A.

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
