# Peer review of "The Status of Wildlife Damage Compensation in China"

_animals, 2024, doi:10.3390/ani14020292_

Round 1
Reviewer 1 Report
Comments and Suggestions for Authors
Overall this is a well-written review article summarizing current legislation and knowledge on the compensation system in China for the human-wildlife conflicts. This work is timely and needed as such information has mostly been available in Chinese only. Authors did an excellent job searching and summarizing what has not been known to the international audience. One key item missing from the manuscript is the conservation status of the species that have caused human-wildlife conflicts. Please add conservation status of each species in supplementary material 2. It is not completely clear if there is a connection between a species’ conservation status and damage compensation. Based on my limited knowledge, China has a quite different hunting law from many countries, making its wildlife protection extend to a very wide range of animals. Authors should consider noting the legal difference in China to help the international audience understanding the unique aspect in human-wildlife conflicts. For example, in most countries hunting wild-boar by the public is a common and effective tool to manage its population. It seems such a management tool might not be available in China. Authors should also consider condensing their recommendations. A few of them seem either vague and not implementable or repetitious. Word choice wise, problem species or harmful species might not be the best term. I recommend nuisance species as it is commonly seen in human-wildlife conflict articles. There are other minor clarifications listed below that might need revision.
Line 77 what did you mean by this study in parentheses.
Somewhere between line 80 to 87 please specify that this review focuses on terrestrial wildlife
Line 125 what is cash crop, please explain
Line 152-154 I am not sure why this sentence is relevant. Only line 155-156 should be sufficient. The only analogy I can think of is how pet or livestock owners are responsible for the damage caused by pet or livestock. Maybe this is what authors tried to say in Line 152-154
Line 159 frequent conflict instead of high conflict?
Line 274-330 it will be helpful if authors can clarify what results came from the literature and what came from their own questionnaire survey.
Line 277 severe instead of serious
Line 301-304 wild boar is not a carnivore. The example does not match the sentence before. Maybe change line 301 carnivore to large mammal?
Line 312 lower numbers compared to what?
Line 379 missing a period after [123]
Line 549-550 Is this a difficulty/challenge or something that should have been done? Did you mean the lack of effective prevention measures?
Line 564 concurrent? Not sure if it is the right word
Line 563-568 this recommendation seems vague. What exact implementation should happen?
Line 570-572 is this recommendation about human casualty
Line 604-607 seems redundant after reading recommendation 1
Line 608-616 It seems 5 and 8 should be combined. This is the scientific research aspect of human-wildlife conflicts, including baseline monitoring, general biology and life history of target organisms, research on damage control methods, risk assessment modeling, etc.
Comments on the Quality of English Languagen/a
Author Response
Overall, this is a well-written review article summarizing current legislation and knowledge on the compensation system in China for the human-wildlife conflicts. This work is timely and needed as such information has mostly been available in Chinese only. Authors did an excellent job searching and summarizing what has not been known to the international audience.
Thank you.
One key item missing from the manuscript is the conservation status of the species that have caused human-wildlife conflicts. Please add conservation status of each species in supplementary material 2. It is not completely clear if there is a connection between a species’ conservation status and damage compensation.
This is an excellent idea and we have added a column stating the conservation status of each species in our supplementary material S1.
Based on my limited knowledge, China has a quite different hunting law from many countries, making its wildlife protection extend to a very wide range of animals. Authors should consider noting the legal difference in China to help the international audience understanding the unique aspect in human-wildlife conflicts. For example, in most countries hunting wild boar by the public is a common and effective tool to manage its population. It seems such a management tool might not be available in China.
We started the second paragraph of the revised version of our Introductions section stating: ‘The Chinese hunting law strictly regulates hunting activities, requiring licenses for legal hunting. The law designates protected species and prohibits hunting during their breeding seasons. Violations result in fines, imprisonment, or confiscation of hunting tools [26, 27]. Chinese conservation efforts to preserve biodiversity and control the im-pact of hunting on wildlife populations, resulted into the effective protection of—often endangered—wildlife species and their habitats [26–28].’
Authors should also consider condensing their recommendations. A few of them seem either vague and not implementable or repetitious.
We followed the reviewer’s recommendation and condensed our recommendations (see our responses to comments below).
Word choice wise, problem species or harmful species might not be the best term. I recommend nuisance species as it is commonly seen in human-wildlife conflict articles.
We followed the reviewer’s recommendation and changed our wording to ‘nuisance species’ throughout the manuscript.
There are other minor clarifications listed below that might need revision.
Line 77 what did you mean by this study in parentheses.
We have omitted the parentheses from the revised version of our manuscript.
Somewhere between line 80 to 87 please specify that this review focuses on terrestrial wildlife.
We have added the information to the revised version of the paragraph: ‘In our article, we therefore focus on the foremost mitigation measure to combat human-wildlife conflict, i.e., the compensation of wildlife damage caused by terrestrial wildlife species, aiming to …’
Line 125 what is cash crop, please explain.
We have changed our wording as follows: ‘Wildlife damage in China refers mainly to 1) damage to local (subsistence) and cash (commercial) crops, stored food, …’
Line 152-154 I am not sure why this sentence is relevant. Only line 155-156 should be sufficient. The only analogy I can think of is how pet or livestock owners are responsible for the damage caused by pet or livestock. Maybe this is what authors tried to say in Line 152-154
We agree and omitted this sentence from the revised version of our manuscript.
Line 159 frequent conflict instead of high conflict?
Omitted together with the previous sentence.
Line 274-330 it will be helpful if authors can clarify what results came from the literature and what came from their own questionnaire survey.
In the method section of the revised version of our manuscript (L119 ff), we clearly state that most information on Hainan Province and Tianjin Municipality presented in our review was obtained from the interview survey. All other information was extracted from the literature research. We further state that information obtained from those surveys is referenced by the citations [48, 49]: ‘Secondly, from June to October 2021, we conducted an interview survey on wildlife damage incidents and subsequent government compensations in Hainan Province and Tianjin municipality. Information obtained from those surveys was previously summarized in two unpublished reports [48, 49]. A total of 3,600 and 45 interview questionnaires were sent to Hainan and Tianjin respectively, while 3,597 and 40 valid responses were returned. Interviewees included were mainly local farmers, livestock keepers, managers and rangers of protected areas or forest plantations, as well as leaders of local forest authorities. We requested information on nuisance species involved in compensation programs, the time, type and seriousness of the damage (Supplementary Material S1). Respondents were also asked to provide information and suggestions on how to improve wildlife damage management and compensation mechanisms.’
Line 277 severe instead of serious
Changed accordingly.
Line 301-304 wild boar is not a carnivore. The example does not match the sentence before. Maybe change line 301 carnivore to large mammal?
We followed the reviewer’s suggestion and changed our wording accordingly: ‘Human-wildlife conflict with large mammals occurs mainly in, or near nature reserves, …’
Line 312 lower numbers compared to what?
We have changed the wording of this sentence as follows: ‘Similarly, to the situation reported by [35] for predators in Gansu and Tibet, the North China leopard (Panthera pardus japonensis) is facing the challenge of decreasing prey numbers in the Tieqiaoshan Provincial Nature Reserve, Shanxi Province. Due to the low numbers of natural prey in the Reserve, leopards were reported to increasingly prey on livestock in and around the protected area, negatively impacting the local communities who entirely depend on livestock rearing as a source of income [38].’
Line 379 missing a period after [123]
Changed accordingly.
Line 549-550 Is this a difficulty/challenge or something that should have been done? Did you mean the lack of effective prevention measures?
Thank you for uncovering this inconsistency. We made this aspect a separate point (6 in the revised version of the manuscript) and changed our wording as follows: ‘Incorporate local ecological knowledge to develop innovative approaches to mitigate human wildlife conflict, e.g., a tree planting initiative in Yunnan Province restored and improved elephant habitat, attempting to keep elephants away from plantations and human settlements [55]. This approach was made possible because the local government was actively enhancing wildlife damage prevention measures and publicly advertising such pro-active approaches.’
Line 564 concurrent? Not sure if it is the right word.
We replaced ‘concurrent with’ by ‘adapt to’.
Line 563-568 this recommendation seems vague. What exact implementation should happen?
The main objective to be implemented is improving and harmonizing management measures and standards for wildlife damage compensation across provinces. We further clarified that compensations should be adapted to the current market price. We do not think that this is vague?!
Line 570-572 is this recommendation about human casualty.
No, it includes all types of damage. We have clarified this in the revised version of this paragraph.
Line 604-607 seems redundant after reading recommendation 1
Following the reviewer’s recommendation, we omitted point 6 of the previous version of the manuscript.
Line 608-616 It seems 5 and 8 should be combined. This is the scientific research aspect of human-wildlife conflicts, including baseline monitoring, general biology and life history of target organisms, research on damage control methods, risk assessment modelling, etc.
In the revised version of our manuscript, we combined the two points (5 and 8): ‘Conduct regular surveys of wildlife damage, to timely recognize potential nuisance species, monitor their population development, activity patterns, behavior, dietary preferences, and key distribution areas [9, 88]. Scientific population control plans for nuisance species that are suitable for population regulation should be formulated and implemented by professional wildlife ecologists, allowing a sustainable control of increasing populations [95]. Culling activities must be carried out by professional, governmental hunters and benefits from culled animals should be transferred to neighboring local communities.’
Reviewer 2 Report
Comments and Suggestions for Authors
Please find comment's attached

Author Response
In this manuscript authors present an overview of the wildlife damage, mitigation, and compensations of it in China. They present a review Chinese legislations and policies regarding wildlife damage compensation, introducing a concept of damage compensation system. Manuscript also analyses 4 case studies in detail. Suggestions on how to improve existing policies and compensation mechanisms in the country are presented. I find this review of great value, as it analyses sources published not only in English, but also in Chinese language (list in national language is presented as Supplement), thus presenting information otherwise not available for the foreign readers. Manuscript deserves publication, my comments below require minor revision of the text, but do not require second review.
Thank you.
General comments
Please stick to the template. Referring, e.g., Line 42: [2–5], not [2-5], go through. Line 44: [8,13,14], not [8, 13, 14], go through. Line 160: [40,58–63], not [40, 58-63], go through.
Changed accordingly throughout the manuscript.
Overuse of numbered lists. Maybe, some of these can be converted to sub-chapters, like on Lines 175, 199?
Only under section 4., we have a third level of sub-heading, i.e. 4.2.1. and 4.2.2. We followed herby the authors instructions and we do not think that this overuse of numbered lists. We therefore kindly refrained from changing the numbering.
As a reader from the different country, I have impression that wildlife damage in China is overestimated, possibly based on the previous system, when no damage was expected, as there were no wild animals doing damage, or their numbers were kept at minimum possible level. If I am right, please introduce in the Problem section by writing additional paragraph. If I am wrong, just make a short note.
Wildlife damage was low in previous decades, but since conservation measures took effect, wildlife numbers are increasing and therefore also the number of human-wildlife conflict as well as the amount of damage caused, and the compensation paid. However, we have the strong feeling that this aspect does not fit into the Difficulties and challenges section, and we therefore refrained from including this statement here. However, we have mentioned this fact earlier in the Introduction section of the revised version of our manuscript.
Title: Even if one of the authors is native speaker, I propose shorter version of the title – “Wildlife damage compensation system in China”. For non-native readers, word “state” means not the “status”, but something related to the country, nation, etc.
We do not think that a title of only eight words is too long. However, we agree that the term ‘state’ might be misleading for non-native English speaker. We have therefore changed the title to: ‘The status of wildlife damage compensation in China.’
Abstract: At least one or two sentences should be given on what was found (e.g., existing compensations are too small) instead on what was done. A good basis might be (1), (5) and (6) from the chapter 7. Difficulties and challenges.
We followed the reviewer’s advise and omitted information on methods and included more information on our findings: ‘We reflect on difficulties and challenges such as delayed damage assessments or compensation quota that do not correspond to current market prices. Since local legislation is often insufficient or completely absent, we make suggestions on how to improve existing policies and compensation mechanisms.’
Methods:
This chapter usually is 2. Materials and Methods
Changed accordingly.
Can you provide a map, where districts analyzed later are shown? Journal is international, and China is a huge country…
We have added a new Figure 1 showing a map of China and highlighting the provinces from which we obtained data and the approximate location of protected areas mentioned in our Case study section (see below), and we give the names of all provinces, autonomous region and municipalities mentioned in the text.
Results:
Chapter 5: it would be of great value to present full list of damage-making species as Supplement.
Our Supplementary Material S1 provides this information already.
Lines 264-273: use as bulleted text, each item from a new line.
Changed accordingly.
6.1 chapter – map would be nice, at least some habitats shown as a basis.
In the revised version of our manuscript, we provided a map showing the locations of the four case study areas.
Figure 1 – mistype in legend and axis
Figure 1 was omitted from our revised manuscript and replaced by the map showing our the locations of our case studies.
6.1 chapter – map would be nice, at least some habitats shown as a basis.
See above.
6.3 chapter – map would be nice, at least some habitats shown as a basis
See above.
6.4 chapter – map would be nice, at least some habitats shown as a basis
See above.
Back matter
Author contributions must be given in different way, check Template
We have adapted the format of the Author contribution section to that suggested by the Journal’s template.
Please list Supplements according to the Template
Following the authors instructions, we indicate the name and title of each element as follows: Table S1: The standards of wildlife damage compensation management in 14 Chinese provinces or autonomous regions, Table S2: Species list of nuisance species, and the type of damage they cause, reported from 19 Chinese provinces, autonomous regions, and municipalities in seven biogeographic regions of China, References S3: Chinese references in Chinese language.
Data Availability Statement: Not applicable
Changed accordingly.
Expand Conflicts of interest as required (role of funders not shown)
We expanded as recommended by the reviewer: ‘The authors declare no conflict of interest. The funding sponsors played no role in the choice of the research project, the design of the study, the collection, analyses or interpretation of data, nor in the writing of the manuscript.’
References
Volume numbers must be in italics
Changed accordingly.
Line 647: page number missing.
This article has no page count.
[17] is whole book and [20] is part of it – I think such citation is wrong
We strongly believe that there is no problem to quote the entire book if the content of the book is relevant to the text, and at the same time quoting only a chapter of that book if the content of only that book section is relevant to the text.
Supplements
Table captions must be presented in every supplement, the same as in the Back Matter, now none present. See Template
Changed accordingly.